# Falls from Great Heights: Risk to Sustain Severe Thoracic and Pelvic Injuries Increases with Height of the Fall

**DOI:** 10.3390/jcm10112307

**Published:** 2021-05-25

**Authors:** Christoph Nau, Maximilian Leiblein, René D. Verboket, Jason A. Hörauf, Ramona Sturm, Ingo Marzi, Philipp Störmann

**Affiliations:** Department of Trauma, Hand, and Reconstructive Surgery, University Hospital Frankfurt, Theodor-Stern-Kai 7, 60590 Frankfurt am Main, Germany; maximilian.leiblein@kgu.de (M.L.); rene.verboket@kgu.de (R.D.V.); jason.hoerauf@kgu.de (J.A.H.); ramona.sturm@kgu.de (R.S.); ingo.marzi@kgu.de (I.M.); philipp.stoermann@kgu.de (P.S.)

**Keywords:** fall, great height, injury pattern, pelvic trauma, spine injury

## Abstract

Falls from a height are a common cause of polytrauma care in Level I Trauma Centers worldwide. The expected injury consequences depend on the height of the fall and the associated acceleration, as well as the condition of the ground. In addition, we further hypothesize a correlation between the cause of the fall, the age of the patient, and the patient’s outcome. A total of 178 trauma patients without age restriction who were treated in our hospital after a fall >3 m within a 5-year period were retrospectively analyzed. The primary objective was a clinically and radiologically quantifiable increase in the severity of injuries after falls from different relevant heights (>3 m, >6 m, and >9 m). The cause of the fall, either accidental or suicidal; age and duration of intensive care unit stay, including duration of ventilation; and total hospital stay were analyzed. Additionally, the frequency of urgent operations, such as, external fixation of fractures or hemi-craniectomies, laboratory parameters; and clinical outcomes were also among the secondary objectives. Sustaining a thoracic trauma or pelvis fractures increases significantly with height, and vital parameters are significantly compromised. We also found significant differences in urgent pre- and in-hospital emergency interventions, as well as organ complications and outcome parameters depending on the fall’s height.

## 1. Introduction

Falls from great heights, either accidentally or intentionally, are common traumatic circumstances in major trauma centers and are often associated with challenging long-term treatment. According to the World Health Organization (WHO), falls are the second-leading cause of accidental injury deaths worldwide, while more than 80% of falls occur in low- and middle-income countries [1]. 

A fall is defined as a downward movement of the body associated with a high kinetic energy, which ends in a sudden vertical deceleration [2]. The height of the fall determines the impact speed and its consequences. Injury patterns depend firstly on whether the body is slowed down during the fall and secondly on the ground conditions [3,4].

They are mostly associated with multiple complex injuries regarding the skeleton and trunk. Depending on the height, life-threatening injuries are more likely to be expected and falls greater than 18 m are usually fatal [5]. Adults older than 65 years of age suffer the greatest number of fatal falls, whereby a high number of the falls are evaluated as so-called ground level falls [1].

Although quite a few studies on falls from heights have been published since the late 1960s, there are only a few that presented specific clinical and laboratory parameters throughout the complete hospital stay. Earlier investigations mostly focused on specific body regions or operative treatments. A few, mostly forensic studies, have presented some age-related differences in injury patterns and influences of height and landing position [4,6]. Other investigations are focused on psychiatric diseases or potential intoxication as an underlying cause of the suicide attempts by falls from great heights [7,8,9]. 

Height affects the expected injuries directly by acceleration and therefore by the landing impact and indirectly by influencing the orientation of the body. Suicidal jumpers tend to sustain more lower extremity fractures, since most of them jump with their feet first [10].

Emergency physicians need to have an understanding of the potential injury patterns in order to be prepared and organized to treat these patients in an often multidisciplinary approach. 

We hypothesize that there is a correlation between the height of the fall, age of the patient, cause of the fall and related injuries and outcome of the patient. One aim of this study is to identify specific injury patterns that are increasingly occurring with greater fall height to further sensitize treatment teams to these injuries in often complex overall injury patterns.

## 2. Materials and Methods

The charts of all patients that were admitted to our university Level 1 Trauma Center located in the city center of a German metropolitan region with suspicion of severe multiple injuries between January 2015 and December 2019 were screened for falls from a great height as the mechanism of injury.

All patients were admitted to our emergency room after a fall from a great height, defined higher than 3 m (ICD-10 code: W09-W17) and were treated as inpatients. All individuals were treated according to the standards of Advanced Trauma Life Support^®^ (ATLS) and received a full body CT scan, as well as additional conventional X-rays, if necessary.

The height of the fall was taken for all patients from the ambulance reports, which were available in all presented cases. Patients were then assigned into three groups depending on their fall height: 3–6.99 m (low), 7–9 m (intermediate), and a fall height of >9 m (high). If the number of floors from which an individual fell were recorded instead of meters, the height of each floor was assumed to be the average value of 3 m based on international standards. The trauma mechanism “fall from a staircase/escalator“ was excluded, independent from the estimated altitude. No age restriction was made, and a sub-analysis of children and adolescents younger than 18 years of age was provided in this study.

Clinical and radiological data were obtained from the patients charts and diagnostic imaging. The cause of the fall, either accidental or suicidal; age and duration of intensive care unit stay, including duration of ventilation; and total hospital stay were analyzed. Additionally, frequency of urgent operations, such as, external fixation of fractures or hemi-craniectomies (Damage Control Surgery: [DCS]); laboratory parameters; and clinical outcomes were also among the secondary variables. Severity of injury was given by the Injury Severity Score (ISS). Base excess (BE), hemoglobin (Hb), and number of thrombocytes were analyzed, and coagulopathy was determined based on partial thromboplastin time (PTT) or INR value, PTT  ≥  40 s or INR  ≥  1.4. 

Patients’ outcome was graduated into 5 categories: good recovery, minor disability (patient independent), severe disability (patient awake but needs support), persistent vegetative state, and dead.

The study was approved by the ethics committee of Frankfurt University Hospital: Grant Number: 20-712. 

### Statistical Analysis

Values are reported as the mean ± standard deviation (SD) for continuous variables and as percentages for categorical variables. All analyses were performed using the Statistical Package for Social Sciences (SPSS for Mac) version 25.0 (IBM Inc., Chicago, IL, USA). Demographic and clinical parameters comparing the different height groups were evaluated using bivariate analysis. *p* values for categorical variables were derived from Chi-square or two-sided Fisher’s exact test and for continuous variables from Student’s t-test or the Mann-Whitney U-test.

## 3. Results

A total of 178 patients whose admission to the clinic was caused by a fall from a height of at least 3 m were analyzed. Overall, 77% (n = 137) of patients were male, and the mean age was 33.7 ± 17.37 years. The mean ISS was 19.58 ± 15.26 points, whereas the thorax (mean Abbreviated Injury Scale [AIS]: 1.67 ± 1.69 points), extremities (AIS: 1.63 ± 1.39 points), and the head (AIS: 1.53 ± 1.74 points) were most severely injured: 74.2% (n = 132) if falls happened accidently, while 25.8% (n = 46) of falls were due to intended suicide. The mean length of intensive care unit stay was 6.41 ± 8.78 days and overall mortality was 4.5% (n = 8) (Table 1).

The patients were then divided into three groups depending on their fall height; 120 patients were assigned to a fall height 3–6 m (low), 36 patients were assigned to a fall height between 7–9 m (intermediate), and 22 patients were assigned to a fall height of > 9 m (high). The mean fall height was 4.4 ± 1.1 m (low), 8.3 ± 0.8 m (intermediate), and 14.1 ± 4.1 m (high) (*p* < 0.001), respectively. In terms of fall height, no differences in terms of gender distribution or age were found (*p* = 0.478; *p* = 0.2). ISS was the highest for patients from the high group (33.77 ± 19.81 points), as compared to that in the intermediate (24.58 ± 15.51 points) and low (15.48 ± 12.03 points) groups, respectively (*p* < 0.001). No differences between the groups were found with regard to the AIS of the head (*p* = 0.738), but significantly higher scores were recorded for the thorax, abdomen, and extremities for patients from the high group (*p* < 0.001) (Table 1). The proportion of attempted suicide was significantly higher in the group of patients from the highest altitude (low: 12.5%, n = 15; intermediate: 44.4%, n = 16; and high: 68.2%, n = 15; *p* < 0.001) (Table 1). Of these patients who attempted suicide, two died in the hospital (4.3%), with one patient (6.3%) in the intermediate and one (6.7%) in the high fall group (no patient from low fall group) (*p* = 0.6).

At hospital arrival, vital parameters, such as systolic blood pressure (SBP, low: 147.26 ± 30.22 mmHg vs. intermediate 141 ± 38.74 mmHg vs. high 123.89 ± 30.33 mmHg, respectively, *p* = 0.017), heart frequency (HF low: 90.56 ± 25.52/min vs. intermediate: 104.69 ± 28.94/min vs. high: 102.33 ± 23.57/min, *p* = 0.01), and the Glasgow Coma Scale (GCS) were lower for (higher for HF) patients from the high group (low: 12.48 ± 4.31 points, intermediate: 10.56 ± 5.26 points, high: 9.73 ± 5.87 points; *p* = 0.014). Furthermore, initially determined Hb values were lower in the “high” group (*p* = 0.004), and the highest BE deviation was present for falls from the highest altitude group (*p* < 0.001) (Table 2).

If they had fallen from the greatest height, patients needed more pre-clinical intubation (low: 9.2% vs. intermediate: 16.7% vs. high: 40.9%, *p* < 0.001) and pre-clinical implantation of chest tubes (low: 1.7% vs. intermediate: 2.8% vs. high: 22.7%, *p* < 0.001), as well as higher ratio of circulatory support by administration of catecholamines (low: 4.2% vs. intermediate: 16.7% vs. high: 22.7%, *p* = 0.004) (Table 3).

In regard to the three groups of different heights, fractures of the pelvis occurred significantly more often if the fall height was at least 6 m (low: 13.4% vs. intermediate: 33.3% vs. high: 45.5%, *p* < 0.001). In addition, chest trauma was more common, the higher the height of the fall was described (low: 41.2% vs. intermediate: 58.3% vs. high: 81.8%, *p* < 0.001). In total, three patients suffered from deceleration of the thoracic aorta (intermediate: 1 vs. high: 2). One of these patients deceased during the initial treatment in the trauma bay. The remaining two patients underwent emergent surgery with implantation of an endovascular prothesis. Traumatic brain injuries were distributed equally between the groups (low: 43.7% vs. intermediate: 41.7% vs. high: 50%, *p* = 0.817) (Table 1), as well as there were no differences with regard to DCS of the head between the groups (low: 10.9% vs. intermediate: 16.7% vs. high: 9.1%, *p* = 0.59) (Table 3). On the contrary, DCS with regard to pelvic trauma was necessary in 18.2% of the “high” group, 8.3% of the “intermediate” group, and 0.8% of the “low” group, respectively (*p* < 0.001). Kinetic therapy due to severe chest trauma was established during the intensive care unit course in 40.9% of the patients from the high group, 25% from the intermediate group, and 11.8% from the low group respectively (*p* = 0.002) (Table 3).

The length of the stay in the intensive care unit was longer if the fall height was higher (low: 4.67 ± 7 days vs. intermediate: 9.72 ± 12.28 days vs. high: 10.5 ± 8.2 days, *p* < 0.001). Nevertheless, no significant differences were detected with regard to the duration of mechanical ventilation (*p* = 0.533). Mortality was higher for the intermediate (11.1%) and high (9.1%) groups when compared to the low group (1.7%, *p* = 0.03) (Table 4).

In this study, 34 (19.1%) patients were younger than 18 years of age. In a subgroup analysis, these patients were further divided into two groups, whether they were younger (n = 15) or older (n = 19) than 6 years of age. Injuries related to body region were comparably distributed between these groups, except for Traumatic Brain Injury (TBI), which were significantly more frequent in the younger patients (n = 12, 80% vs. n = 8, 42.1.%; *p* = 0.026). All falls from a height in children under 6 years of age were accidental, whereas five children from the older group jumped from a height intentionally (*p* = 0.031), where the youngest patient was 14 years old.

## 4. Discussion

Falling from a great height is a common reason for major trauma and is associated with high mortality and high overall injury severity [11]. For this reason, falls from great heights account for one of the trauma mechanisms where there is a recommendation to treat patients in specialized trauma centers. For example, the level 3 guideline of the German Society for Trauma Surgery (DGU) recommends trauma team activation for fall heights greater than 3 m [12], which is in line with the guidelines of the American College of Surgeons. Demetriades et al. reported that 2.3% of all traumatological hospital admissions were due to falls from a great height (defined as higher than 15 feet [4.6 m]) [13]. In this regard, Simmons et al. established a fall height of more than 20 feet (6.1 m) as a predictor for the presence of major trauma (ISS > 15 points) [14]. A significant relationship between a higher ISS, a worse GCS, and the need for transfusion was described by Alizo et al. for heights equaling at least 25 feet (7.6 m), recently [15]. Despite its enormous impact in trauma care, data with regard to the suspected injury pattern caused by different heights of falls remains sparse. Therefore, in this study we tried to identify injury patterns with regard to the height of the fall to get more insight into the injury pattern. In our study, TBIs, as well as spine injuries and the associated emergency surgical interventions were distributed equally across groups, regardless of the altitude. According to these results, Giordano et al. reported about only 15.3% spine injuries in falls from great heights [16]. In contrast, we found an increasing number of pelvic fractures, extremity injuries, and thoracic injuries with increasing altitude. This is similar to previously published results that demonstrated an increase in pelvic fractures, as well as more thoracic injuries requiring surgical treatment with greater fall height [15,16,17,18]. Furthermore, the abdomen and extremities are affected to a significantly greater extent when correlated to the height of the fall. These results are in line with previously published results, where the incidence of chest injuries and pelvic fractures increased by about 30% if the fall height was estimated to be more than 7 m [3]. Unstable pelvic ring fractures are typical for high-energy trauma and, in addition to traffic accidents, are caused in large numbers by falls from great heights, which emphasizes our findings [19,20].

In addition to these special features in regard to the injury distribution, we found higher mortality in falls from greater heights. Some correlations between injury severity and mortality with regard to the height have been described in the literature before [15,21]. For example, a height of more than 14.6 m is considered to be associated with a mortality rate above 50%, while almost all falls above a height of 18.6 m are considered to be fatal [5]. A correlation between greater fall height and injury severity was shown by Fang et al., while Giordano et al. described increasing mortality from a fall height above 8 m [16,21]. Simmons et al. described a fall from a height of at least 20 feet (6.1 m) as the threshold for suffering from major trauma [14].

Besides the actual altitude of the fall, the distribution of injuries and the ISS depends on several other factors, such as the landing surface and the patients position upon landing [18,22,23,24]. For example, the absorption of the direct impact by the pelvis with a high number of injuries is conceivable as an explanation for the relatively low number of vertebral fractures due to the reduced kinetic energy that is transmitted to other body regions when primarily absorbed by the pelvis [22]. Another explanation for the lack of increase in incidence or severity of TBI may be that nearly 70% of prehospital deaths are attributed to the presence of severe TBI at the scene; thus, these cases no longer reach the hospital [16].

A large subgroup of patients falling from a great height consists of patients who attempt to commit suicide. In retrospective forensic studies, the authors described thoracic and craniofacial fractures being more likely in intentional jumps rather than in accidental falls and they also found a correlation between a non-deformable landing surface and the fracture pattern, resulting more often in a combination of axial and appendicular fractures, especially pelvic and lower extremity fractures [25,26].

Casali et al. analyzed 307 deaths after jumps from great heights. In the deceased, injuries of not only the lung parenchyma, liver, and kidney, but also of the thoracic aorta were found to a large extent. Interestingly, the mentioned study was able to show a correlation between fall height and the presence of bony injuries of the pelvis, which is consistent with the results of our study [27]. Since the rate of suicide attempts rise with the altitude in our collective, it can be assumed that jumpers deliberately choose greater heights. Interestingly, Piazzalunga et al. reported that the AIS of the extremities in suicidal jumpers was high, confirming a “feet first landing” in these patients. Furthermore, in-hospital mortality in this group was comparable to that found in our (total) collective (4.7%) [28]. Borg et al. found that jumping from high altitudes accounts for about 5% of all suicide attempts in Sweden [1]. Nevertheless, an estimated number of unknown cases in this group must be assumed, since a large number of jumps from a greater height than nine meters are not survived and therefore do not present to emergency departments. Nevertheless, especially in low- and middle-income countries, falls make the second leading cause of death, with an estimated 424,000 deaths per year due to falls, followed by road traffic accidents [2].

Severe pediatric trauma is associated with numerous peculiarities, both in the injury pattern and in the treatment. For children, falls are the most common reason to suffer unintended injuries and the second leading cause of injury-related deaths [29]. Over 7 years in the US, 1061 deaths of patients younger than 18 years of age resulted from falls from great heights [30]. In this study, 17.3% of all patients suffered from multiple trauma and 63.2% presented with head trauma [30]. In our study, 34 patients younger than 18 years of age were included. In comparison to adults, falls in this group mainly occurred accidentally. Regarding the injury pattern, we found significantly more TBI when compared to the adult patients in our collective, which once more emphasizes the need for special care in these critically injured children in specialized centers. Randazzo et al. examined 47,351 minors who fell from a tree house, of which 29.3% fell from a height of more than 10 feet (3.05 m) and were thus well comparable to patients from the present study. For children younger than five years of age, the odds of sustaining a head injury were higher, which is in line with our findings and pediatric trauma in general. Kafadar et al. analyzed pediatric falls in 1326 patients, of whom 13.2% fell from a height greater than 5 m [31]. The proportion of patients younger than four years was highest, causing authors to conclude that falls from great heights in children are most likely to occur from dangerous and unmonitored play situations.

The overall lethality in our study was quite low when compared to the overall in-house lethality of severely injured patients in Germany (4.5 vs. 11%) in the TR-DGU^®^ report 2020. One might assume that due to the high impact caused by falls from those great heights, a relevant number of patients had already passed away preclinically and therefore were not admitted to any hospital. Additionally, lethality was significantly lower in the low group in our study, which once more emphasizes the relationship between the height of the fall and lethality.

The results of this study extend the knowledge from currently available literature. With increasing fall height, the overall injury severity is higher. This appears to be due to a significantly higher severity of thoracic and pelvic injuries, whereas the severity of head injuries and fractures of the spine were not significantly more severe as a function of fall height. As a consequence, emergency stabilization of the pelvis had to be performed significantly more often in the high fall group, which emphasizes the requirement to “clear the pelvis” in patients jumping from the greatest height.

## 5. Limitations

This study has some limitations. First of all, its retrospective character with all accompanying restrictions due to the study design. Thus, the data about the height of the fall could be taken from the rescue service protocols, and in most cases, no valid information was available about the nature of the ground—which must be considered when interpreting the results. Furthermore, it can be assumed that the rate of prehospital deaths is relatively high, especially with increasing height. Against this background, the data on lethality should be interpreted with caution, as there is a clear selection bias due to a presumably high rate of preclinical patient deaths. Therefore, this study does not provide any information on injuries that directly cause death.

## Figures and Tables

**Table 1 jcm-10-02307-t001:** Basic patient data in regard to the height of the fall. (ISS: Injury Severity Score, AIS: Abbreviated Injury Scale).

	Total (n = 178)	3–6 m (n = 120)	7–9 m (n = 36)	>9 m (n = 22)	*p*-Value
Height of fall (m)	6.4 ± 3.7	4.4 ± 1.1	8.3 ± 0.8	14.1 ± 4.1	< 0.001
Male sex (%, n)	77 (137)	79.2 (95)	69.4 (25)	77.3 (17)	0.478
Age (years)	33.7 ± 17.37	35.23 ± 17.67	28.94 ± 17.53	33.14 ± 14.52	0.2
Attempted suicide (%, n)	25.8 (46)	12.5 (15)	44.4 (16)	68.2 (15)	< 0.001
Accident (%, n)	74.2 (132)	87.5 (105)	55.6 (20)	31.8 (7)	< 0.001
ISS (points)	19.58 ± 15.26	15.48 ± 12.03	24.58 ± 15.51	33.77 ± 19.81	< 0.001
AIS_head_ (points)	1.53 ± 1.74	1.48 ± 1.67	1.78 ± 1.97	1.36 ± 1.76	0.738
AIS_face_ (points)	0.35 ± 0.83	0.42 ± 0.9	0.28 ± 0.74	0.14 ± 0.47	0.363
AIS_thorax_ (points)	1.67 ± 1.69	1.29 ± 1.55	2.11 ± 1.58	3.05 ± 1.76	< 0.001
AIS_abdomen_ (points)	1.18 ± 1.52	0.86 ± 1.27	1.56 ± 1.58	2.32 ± 1.99	< 0.001
AIS_extremities_ (points)	1.63 ± 1.39	1.34 ± 1.25	2 ± 1.43	2.64 ± 1.47	< 0.001
AIS_softtissue_ (points)	0.29 ± 0.6	0.28 ± 0.52	0.33 ± 0.72	0.32 ± 0.78	0.868
Traumatic brain injury (%, n)	44.1 (78)	43.7 (52)	41.7 (15)	50 (11)	0.817
Thoracic trauma (%, n)	49.7 (88)	41.2 (49)	58.3 (21)	81.8 (18)	0.001
Fracture of upper extremity (%, n)	32.2 (57)	29.4 (35)	33.3 (12)	45.5 (10)	0.33
Fracture of lower extremity (%, n)	31.1 (55)	26.1 (31)	41.7 (15)	40.9 (9)	0.118
Fracture of the pelvis (%, n)	21.5 (38)	13.4 (16)	33.3 (12)	45.5 (10)	0.001
Fracture of the spine (%, n)	38.4 (68)	36.1 (43)	44.4 (16)	40.9 (9)	0.646

**Table 2 jcm-10-02307-t002:** Vital parameters and laboratory parameters at admission sorted by the altitude of the fall.

	Total (n = 178)	3–6 m (n = 120)	7–9 m (n = 36)	>9 m (n = 22)	*p*-Value
Systolic blood pressure (mmHg)	142.99 ± 32.84	147.26 ± 30.22	141 ± 38.74	123.89 ± 30.33	0.017
Heart frequency (beats per minute)	95.07 ± 26.61	90.56 ± 25.52	104.69 ± 28.94	102.33 ± 23.57	0.01
Glasgow Coma Scale in-hospital (points)	11.75 ± 4.82	12.48 ± 4.31	10.56 ± 5.26	9.73 ± 5.87	0.014
Hemoglobin (g/dl)	12.81 ± 2.42	13.28 ± 1.98	12.09 ± 3.02	11.44 ± 2.78	0.004
Platelet count (/nl)	230,493 ± 81,286	235,050 ± 73,398	228,911 ± 100,648	208,636 ± 86,843	0.214
Partial thromboplastin time (seconds)	27.75 ± 8.04	26.71 ± 4.35	30.03 ± 13.9	29.67 ± 9.91	0.308
Quick (%)	90.22 ± 17.58	91.56 ± 17.14	90.46 ± 17.91	82.76 ± 18.34	0.164
International normalized ratio	1.1 ± 0.18	1.09 ± 0.16	1.12 ± 0.24	1.14 ± 0.16	0.424
Base excess venous (mmol/l)	−2.85 ± 5.66	−1.05 ± 3.81	−6.15 ± 7.31	−7.05 ± 6.37	< 0.001

**Table 3 jcm-10-02307-t003:** Urgent pre- and in-hospital emergency interventions. (ETI: endotrachial intubation, CPR: cardio pulmonary resuscitation, DCS: Damage Control Surgery).

	Total (n = 178)	3–6 m (n = 120)	7–9 m (n = 36)	>9 m (n = 22)	*p*-Value
Pre-clinical ETI (%, n)	14.6 (26)	9.2 (11)	16.7 (6)	40.9 (9)	0.001
Pre-clinical CPR (%, n)	3.4 (6)	0.8 (1)	8.3 (3)	9.1 (2)	0.026
Pre-clinical catecholamines (%, n)	9 (16)	5.8 (7)	11.1 (4)	22.7 (5)	0.034
Pre-clinical chest tube (%, n)	4.5 (8)	1.7 (2)	2.8 (1)	22.7 (5)	< 0.001
In-hospital CPR (%, n)	1.1 (2)	0	0	9.1 (2)	0.001
In-hospital catecholamines (%, n)	9 (16)	4.2 (5)	16.7 (6)	22.7 (5)	0.004
In-hospital chest tube (%, n)	12.9 (23)	8.3 (10)	19.4 (7)	27.3 (6)	0.022
DCS brain (%, n)	11.9 (21)	10.9 (13)	16.7 (6)	9.1 (2)	0.59
DCS upper extremity (%, n)	11.3 (20)	9.2 (11)	13.9 (5)	18.2 (4)	0.41
DCS lower extremity (%, n)	18.2 (32)	14.4 (17)	30.6 (11)	18.2 (4)	0.089
DCS pelvis (%, n)	4.5 (8)	0.8 (1)	8.3 (3)	18.2 (4)	0.001
DCS spine (%, n)	13 (23)	11.8 (14)	13.9 (5)	18.2 (4)	0.702
Dorso-ventral stabilization (%, n)	6.8 (12)	5.9 (7)	5.6 (2)	13.6 (3)	0.392
Kinetic therapy (%, n)	18.1 (32)	11.8 (14)	25 (9)	40.9 (9)	0.002

**Table 4 jcm-10-02307-t004:** Organ complications and outcome parameters. (CNS: central nerve system, ICU: Intensive Care Unit).

	Total (n = 178)	3–6 m (n = 120)	7–9 m (n = 36)	>9 m (n = 22)	*p*-Value
Organ failure lung (%, n)	14 (25)	10.8 (13)	13.9 (5)	31.8 (7)	0.034
Organ failure circulation (%, n)	20.2 (36)	12.5 (15)	33.3 (12)	40.9 (9)	0.001
Organ failure cns (%, n)	19.7 (35)	15.8 (19)	25 (9)	31.8 (7)	0.148
Mechanical ventilation (%, n)	59 (69)	54.5 (36)	61.3 (19)	70 (14)	0.447
Length of ICU stay (days)	6.41 ± 8.78	4.67 ± 7	9.72 ± 12.28	10.5 ± 8.2	<0.001
Length of mechanical ventilation (days)	3.86 ± 7.12	3.22 ± 5.04	4.87 ± 11.04	4.4 ± 5.2	0.533
Mortality (%, n)	4.5 (8)	1.7 (2)	11.1 (4)	9.1 (2)	0.03

## Data Availability

Data is contained within the article.

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
