# Peer review of "Falls from Great Heights: Risk to Sustain Severe Thoracic and Pelvic Injuries Increases with Height of the Fall"

_jcm, 2021, doi:10.3390/jcm10112307_

Round 1

Reviewer 1 Report

Dear authors,

thank you very much for your paper.

Overall, I´m missing the key message. What can I learn from you? The higher the fall, the more severe the injury - This can not be everything.

Comments:

The grouping is very weak - you describe this (not very sufficient) in the limitation section. It must be clearly shown in the method section that the heights are preclinically estimated!

Therefore, it is not intelligible why the drop height in Tbl 1 is given to the 2nd decimal place.

The labeling of all tables for the 3rd column (drop height) is wrong. You write 7-9 m, it should read 6-9 m (precise: [3;6[ m, [6;9[ m, >= 9 m, or [3;6] m, ]6;9] m, >9m). In which groups are the 6 m and 9 m drop heights? Unfortunately, this remains unclear.

In Table 2, you report a GCS of 9.73 for the third group. This value is improbable. With 9 intubated patients (GCS = 3) in the group, the maximum possible mean is 10.09.  

Table 3 is not present, shown is Table 4.

An evaluation of aortic dissection would be very interesting. With a total population of 178, this entity should still be post-evaluable.

Line 156: Please reference Tbl. 1

Line 236 ff: "...the present study was able .... which is consistent with the results of our study." This is misleading. The present study is usually your study.

Line 243: Please calculate the lethality of suicides of your population! 

Line 272: You mean lethality, not mortality! An adjusted lethality (e.g., SMR) would be much more meaningful.

Thank you.

Reviewer 2 Report

Thank you for the opportunity to review this manuscript. This paper presents a retrospective review of the injuries that result from falls of >3m. Although a review of injuries from high falls is not novel, this study further adds to the body of literature on this topic. The study is well done and is appropriate for publication; the methodology is sound, the results are interesting, and the discussion appropriately addresses the key findings. The manuscript does, however, require minor revisions before it may be considered suitable for publication. With all scholarly respect, please consider the following recommendations/edits.

Introduction

  • The authors state “only a few studies have been published”. I would disagree. There has been extensive research in clinical medicine since the 1960s documenting the injuries that result from high falls. Please engage with some of this literature to better highlight the gap in knowledge that this study is augmenting.
  • The aim of the paper is not clear. Please clearly articulate the aim of the research before presenting the hypothesis.

Materials & Methods

  • Please provide the details of the hospital rather than saying ‘our’
  • In regard to terminology, please note that ‘mechanism’ refers to the force that causes injury to the body. So, in the case of falls, the mechanism is blunt force. The type of fall, in this case falls from heights, is considered the traumatic event/circumstances.
  • The fourth paragraph requires clarification and re-wording. As I read it, the first sentence of this paragraph (“The primary objective …”) actually reads as the aim of the paper. That is, the aim of the study was to use clinical and radiology data to examine if injuries increase in severity as the height of the fall increases. At the moment, this sentence needs re-wording, due to poor English expression, and should be moved to the ‘Introduction - aim’ section of the paper. For the rest of this paragraph, may I suggest you use the word ‘variable’, rather than ‘objective’, when describing the information you are collecting. The word objective implies an aim. So for this paragraph, may I suggest the first couple of sentences are re-worded to clearly say what variables were collected from the clinical information and what variables were collected from radiology information.

Discussion

  • Please re-word the discussion on mortality within the framework of selection bias. As the mortality data is this study is incomplete (i.e., not a true representation of all the falls that resulted in death), and you do not have a true snapshot of the mortality figures, I don’t think the discussion on workplace safety is justified.

The paper would benefit from a short concluding paragraph summarising the key findings and how they address the research aim.

English expression throughout the manuscript (including abstract) requires improvement. The poor English expression does, at times, hinder the flow of the paper and allows for misinterpretation of what the authors are saying. Further, the use of commas needs to be improved; sentences should always start with a word, and there are several typos throughout. Please also do not treat single sentences as complete paragraphs.

Round 2

Reviewer 1 Report

Thank you for your detailed answer and the corrections.

The labeling of the tables is unfortunately misleading and mathematically false. But I understand now how you made the grouping. Normally, the grouping is shown in materials and methods. Please write a sentence in materials and methods that group 1 is 3-6.99m and group 2 is 7-9m and group 3 > 9m.

Tank you for checking the GCS of 9.73 in group 3.

For your understanding: Nearly all nonintubated patients in the group 3 had to have a GCS of 15 in the shock room to achieve this score of 9.73 (maximum possible mean: (9x3+13x15)/22=10.09). I am amazed at a GCS of 15 after a fall >9 m with so many people.

Thank you, I'm looking forward to the release.
